# CRISPR-Cas9-Mediated Correction of *SLC12A3* Gene Mutation Rescues the Gitelman’s Disease Phenotype in a Patient-Derived Kidney Organoid System

**DOI:** 10.3390/ijms24033019

**Published:** 2023-02-03

**Authors:** Sun Woo Lim, Xianying Fang, Sheng Cui, Hanbi Lee, Yoo Jin Shin, Eun Jeong Ko, Kang In Lee, Jae Young Lee, Byung Ha Chung, Chul Woo Yang

**Affiliations:** 1Transplantation Research Centre College of Medicine, The Catholic University of Korea, Seoul 06591, Republic of Korea; 2Department of Internal Medicine, Division of Nephrology, Seoul St. Mary’s Hospital, College of Medicine, The Catholic University of Korea, Seoul 06591, Republic of Korea; 3R&D Center, ToolGen, Inc., Seoul 06591, Republic of Korea

**Keywords:** Gitelman’s disease, CRISPR/Cas9, induced pluripotent stem cells, kidney organoid

## Abstract

The aim of this study is to explore the possibility of modeling Gitelman’s disease (GIT) with human-induced pluripotent stem cell (hiPSC)-derived kidney organoids and to test whether gene correction using CRISPR/Cas9 can rescue the disease phenotype of GIT. To model GIT, we used the hiPSC line CMCi002 (CMC-GIT-001), generated using PBMCs from GIT patients with *SLC12A3* gene mutation. Using the CRISPR-Cas9 system, we corrected CMC-GIT-001 mutations and hence generated CMC-GIT-001^corr^. Both hiPSCs were differentiated into kidney organoids, and we analyzed the GIT phenotype. The number of matured kidney organoids from the CMC-GIT-001^corr^ group was significantly higher, 3.3-fold, than that of the CMC-GIT-001 group (12.2 ± 0.7/cm^2^ vs. 3.7 ± 0.2/cm^2^, *p* < 0.05). In qRT-PCR, performed using harvested kidney organoids, relative sodium chloride cotransporter (NCCT) mRNA levels (normalized to each iPSC) were increased in the CMC-GIT-001^corr^ group compared with the CMC-GIT-001 group (4.1 ± 0.8 vs. 2.5 ± 0.2, *p* < 0.05). Consistently, immunoblot analysis revealed increased levels of NCCT protein, in addition to other tubular proteins markers, such as LTL and ECAD, in the CMC-GIT-001^corr^ group compared to the CMC-GIT-001 group. Furthermore, we found that increased immunoreactivity of NCCT in the CMC-GIT-001^corr^ group was colocalized with ECAD (a distal tubule marker) using confocal microscopy. Kidney organoids from GIT patient-derived iPSC recapitulated the Gitelman’s disease phenotype, and correction of *SLC12A3* mutation utilizing CRISPR-Cas9 technology provided therapeutic insight.

## 1. Introduction

Gitelman’s disease (GIT) is a genetic tubular disorder with an autosomal recessive inheritance pattern. Mutations in the solute carrier family 12 member 3 (*SLC12A3*) gene, which encodes the sodium chloride cotransporter (NCCT) [1], also known as the thiazide sensitive cotransporter, underlie this disease. NCCT expresses itself in the apical membrane of cells in the first part of the distal convoluted tubule (DCT). In health, the DCT is responsible for 5–10% of renal sodium reabsorption [2]. In patients with GIT, less sodium is reabsorbed in the DCT due to defective NCCT; hence, the increased sodium delivery to the collecting duct causes distal sodium rescue. As a result, potassium and hydrogen excretion increase. Therefore, the characteristic biochemical picture of hypokalemic alkalosis occurs [3].

Although the genetic and molecular pathophysiology of GIT has been established, the current treatment options for GIT are very limited. Lifelong replacement of high potassium, magnesium, and sodium, along with diet modification, remains the mainstay of therapeutic strategies. However, even with a delicate prescription for electrolyte supplementation and the advice of a diet specialist regarding treatment, it can be difficult to correct electrolyte abnormalities, and severe electrolyte imbalance, which needs urgent medical therapy, can occur. Therefore, an effective treatment option for GIT has not existed till now, and an ultimate therapeutic strategy is required.

Meanwhile, the development of CRISPR/Cas9 technology has enabled rapid and efficient gene editing [4]. One of the fastest growing applications of CRISPR/Cas9 approaches is in gene therapy to cure genetic diseases. Correcting genetic mutations in situ using an appropriate model, such as a kidney organoid system, is a promising approach to genetic diseases for which no effective treatment methods exist, such as GIT [5]. Indeed, some previous studies have already generated isogenic hiPSCs (human-induced pluripotent stem cells) with repaired gene mutations and shown that disease phenotypes have been corrected as well [6]. In kidney disease, there have been attempts to apply the CRISPR/Cas9 system to remove genes associated with porcine endogenous retroviruses to enable kidney xeno-transplantation or ex vivo correction of mutation for autosomal dominant polycystic kidney disease [5]. However, the modeling of GIT using a kidney organoid system has never been attempted with the application of CRISPR/Cas9 gene correction until now.

Previously, we have generated hiPSCs (CMC-GIT-001) using peripheral blood mononuclear cells (PBMCs) from patients diagnosed with GIT through *SLC12A3* gene sequencing [7]. In this study, we attempted to model GIT by differentiating CMC-GIT-001 into kidney organoids using a well-established differentiation protocol and confirmed whether it can represent the phenotype of GIT. Next, we planned to correct the *SLC12A3* gene mutation found in CMC-GIT-001 and to also investigate whether NCCT expression can be restored, in order to appraise the possibility of applying gene therapy in the treatment of GIT.

## 2. Results

### 2.1. Targeted Gene Correction of SLC12A3 Mutation with the CRISPR/Cas9 System

DNA sequencing shows heterozygous *SLC12A3* gene mutations with the deletion of T and G nucleotides (c.46_47del) in exon 1 and T to C (c.2963T>C) mutation in exon 26, which both cause amino acid mutations, Cys16Glnfs*13 and Ile988Thr, respectively (Figure 1A). We aimed to generate isogenic wild-type hiPSC lines using targeted gene correction to rescue the disease phenotype of GIT using the CRISPR/Cas9 system and single-stranded oligodeoxynucleotides (ssODNs).

We first designed an exogenous ssODN template containing the wild-type (WT) coding sequence and homonyms arms spanning the target site as the repair template and single guide RNA (sgRNA) to recognize DNA, bind the endonuclease, and induce site-specific cleavage (Figure 1B) [8]. To define the sgRNAs that can be recognized by the CRISPR/Cas9 enzyme at the *SLC12A3* gene sites, we inserted each exon region into CRISPR RNA-guided endonuclease (RGEN) tools. Figure 1C shows that selected sgRNAs have 0-0-0 for exon 1 mismatches, which indicates lower probabilities of off-target reactivity [9]. An in vitro Cas9 cleavage assay was performed to test the efficiency of the sgRNAs using genomic DNA from patient hiPSCs. As shown in Figure 1D, we observed fragmented two band (marked red stars) in the treatment of both Cas9 nuclease and sgRNA relative to the positive.

We next evaluated the patient hiPSC line with *SLC12A3* correction, CMC-GIT-001^corr^ hiPSC. The *AccIII* restriction fragment length polymorphism was specific to the corrected allele, and DNA sequencing also confirmed that *SLC12A3*^+/−^ was correctly targeted without any footprint. As shown in Figure 1E, PCR products using primer 1 and 2 for exon 1 were digested by the *AccIII* enzyme into 294 bp and 231 bp fragments for exon 1, respectively. In addition, we also checked the DNA sequence of the generated hiPSC lines following targeted DNA sequencing (Figure 1F).

### 2.2. Establishment of GIT Patient^corr^ hiPSCs

The generated CMC-GIT-001^corr^ hiPSCs displayed typical pluripotent stem cell-like morphology and expressed pluripotent markers, such as NANOG, SSEA-4, and TRA-1-81, according to flow cytometry and immunofluorescence (Figure 2A–C). Tri-lineage differentiation assays demonstrated that CMC-GIT-001^corr^ hiPSCs were successfully differentiated into ectoderm, mesoderm, and endoderm (Figure 2D). CMC-GIT-001^corr^ hiPSCs maintained normal karyotypes after CRISPR/Cas9 editing (Figure 2E). The absence of mycoplasma contamination in the generated hiPSC line was shown with a standardized PCR test, as shown in Figure 2F.

### 2.3. Differentiation of Corrected CMC-GIT-001 into Kidney Organoids

In order to model GIT within the context of the kidney, we differentiated both CMC-GIT-001 and CMC-GIT-001^corr^ hiPSC lines into kidney organoids using a previously established adherent culture protocol (Figure 3A) [10,11,12]. A total of 21 days after plating, typical segmented tubular structures were detected. However, we observed delayed kidney organoid formation upon whole-well bright field observation in the CMC-GIT-001 compared with CMC-GIT-001^corr^, as shown in Figure 3B.

Next, we counted the number of matured kidney organoids based on the features that WTC-11 hiPSC-derived kidney organoids express, being a nephron marker and size greater than 100 μm. The number of matured kidney organoids from CMC-GIT-001 hiPSCs was markedly lower than that of WTC-11 (3.7 ± 0.2/cm^2^ vs. 16.7 ± 1.3/cm^2^, *p* < 0.05 vs. WTC-11 group). However, we found that the number of matured kidney organoids was recovered in the CMC-GIT-001^corr^ hiPSC-derived kidney organoids compared to that of CMC-GIT-001 hiPSCs (12.2 ± 0.7/cm^2^ vs. 3.7 ± 0.2/cm^2^, *p* < 0.05) (Figure 3C). Upon quantitative analysis of the number of matured kidney organoids, it was confirmed that delayed kidney organoid differentiation of CMC-GIT-001 hiPSC significantly could be recovered by genetic engineering of *SLC12A3* of CMC-GIT-001 using CRISPR/Cas9.

Immunofluorescence staining and examination under confocal microscopy revealed that WTC-11 as wild-type (WT) was successfully differentiated into kidney organoids without major structural differences, expressing markers of nephron structure, including PODXL in glomerular epithelial cells, LTL in the proximal tubule, and E-cadherin (ECAD) in the distal tubule, in appropriately patterned and contiguous segments as shown in high magnification in Figure 3D. However, the kidney organoids of GIT patient hiPSCs did not show the typical structure seen in control organoids. In contrast, the generated CMC-GIT-001^corr^ hiPSCs differentiated into kidney organoids with normal nephron marker expression and patterns, as shown in Figure 3D.

### 2.4. Recovery of NCCT Expression in Kidney Organoids from CMC-GIT-001^corr^ hiPSCs

To confirm the targeted gene correction of *SLC12A3*, we evaluated the expression of NCCT in kidney organoids derived from CMC-GIT-001^corr^ hiPSCs. Using qRT-PCR, NCCT mRNA levels were significantly decreased in CMC-GIT-001 compared to WTC-11-derived organoids (33 ± 4.8% vs. 100 ± 7%, *p* < 0.05). However, this level was recovered in CMC-GIT-001^corr^, as shown in Figure 4A (100 ± 19% vs. WTC-11 group or CMC-GIT-001 group). Consistent with these results, reduced NCCT protein levels in CMC-GIT-001-compared with WTC-11-derived organoids (30 ± 3.8% vs. 100 ± 5%, *p* < 0.05) also significantly increased in the CMC-GIT-001^corr^ compared to the CMC-GIT-001 (58 ± 3.2%, *p* < 0.05 vs. CMC-GIT-001 group) (Figure 4B,C). The results of molecular analysis of NCCT using qRT-PCR and immunoblot demonstrated that gene correction using CRISPR/Cas9 rescued NCCT mRNA and protein expression in CMC-GIT-001 hiPSC with mutation.

We next performed double immunolabeling with NCCT and ECAD to determine the localization and expression level of NCCT, which is mainly localized to the apical membrane of ECAD-positive tubular cells in WTC-11 hiPSC kidney organoids. However, the immature kidney organoids derived from CMC-GIT-001 hiPSC did not react with the antibodies against both NCCT and ECAD. On the other hand, the kidney organoids from CMC-GIT-00^corr^ 1 restored the expression of NCCT as well as ECAD compared to those of CMC-GIT-001, as shown in Figure 4D.

We next compared the protein expression of the other nephron markers, LTL and ECAD, in the kidney organoids from WTC-11 hiPSC, CMC-GIT-001, and CMC-GIT-001^corr^ hiPSCs. Consistent with the immunofluorescence staining results, reduced LTL (25 ± 10% vs. 100 ± 8%, *p* < 0.05) and ECAD (52 ± 10% vs. 100 ± 4%, *p* < 0.05) protein levels in the CMC-GIT-001 compared with WTC-11 significantly recovered in CMC-GIT-001^corr^ organoids (LTL, 62 ± 8.5%, *p* < 0.05 vs. CMC-GIT-001 group; ECAD, 69 ± 4.8%, *p* < 0.05 vs. CMC-GIT-001 group), as shown in Figure 5A,B. These results suggest that that reduced expression of LTL and ECAD in the kidney organoid from CMC-GIT-001 hiPSC is related to the retarded differentiation, but CMC-GIT-001^corr^ generated by CRISPR/Cas9-mediated gene correction significantly restored their expression as well as morphological maturation.

## 3. Discussion

In this study, we used CRISPR/Cas9 in GIT patient-derived hiPSCs to correct the mutation responsible for disease development and differentiated the corrected hiPSCs into kidney organoids. As a result, in the gene-corrected kidney organoids, the structure showed a more mature form and expression representative of renal tubular cells, demonstrating rescue of the GIT disease phenotype.

First, we used the previously reported patient-derived hiPSCs as a source to model GIT with a kidney organoid and for the application of CRISPR/Cas9 [7]. In previous reports, kidney organoids generated using hiPSCs contained segmented structures with podocytes, proximal tubules, and distal tubules in nephron-like arrangements and also showed native genomic context and cell-type heterogeneity within the kidney [12,13,14]. In addition, kidney organoids from genetically modified or patient-derived hiPSCs can successfully recapitulate the phenotype of various genetic kidney diseases [10,12,14,15,16]. Regarding GIT, defects in NCCT are the hallmark of disease phenotype, so we first compared the expression of NCCT between WT and CMC-GIT-001. We found pre-dominant expression of NCCT in the apical part of the cells along with E-cadherin, the marker for distal tubular epithelial cells, in WT kidney organoids, which is consistent with previous knowledge [17,18]. In contrast, the expression of NCCT was significantly decreased in IF staining for NCCT in CMC-GIT-001 kidney organoids compared to WT kidney organoids, confirmed in protein and RNA quantitative analysis for NCCT. These findings suggest that modeling GIT using patient-derived kidney organoids was successful in this study.

Next, we tried to correct for *SLC12A3* gene mutation using CRISPR/Cas9. Some previous studies already suggest that CRISPR/Cas9-mediated gene correction has potential as a therapeutic strategy for genetic diseases, especially when an appropriate treatment strategy is unavailable, such as GIT [15,19]. For example, correction of mutation in hereditary hematologic disorders, such as ß-thalassemia or hemophilia A, using patient-derived hiPSCs rescues the disease phenotype [20,21]. In another study, CRISPR/Cas9-mediated correction was performed in dominant optic atrophy in patient-derived hiPSCs [22]. In this study, CMC-GIT-001 showed heterozygote mutations in two different sites reported as pathogenic for the development of GIT according to ACMG guidelines. After gene correction by CRISPR/Cas9, mutation of c. 47_48del in exon 1 was successfully corrected, and mRNA and protein of NCCT were also significantly recovered, according to qRT-PCR and immunoblot analysis. This finding suggests the successful rescue of the GIT disease phenotype through CRISPR/Cas9. However, we found that NCCT, the expression of mRNA and protein recovered by 3-fold and 2-fold respectively compared with CMC-GIT-001 according to qRT-PCR and immunoblot analysis. It is necessary to investigate the periodic expression of mRNA and protein of NCCT during kidney organoid differentiation in the future.

Interestingly, the proportion of mature kidney organoids was significantly low, and tubular proteins, such as LTL and ECAD, showed decreased expression, and PODXL was not detected in CMC-GIT-001 kidney organoids in comparison with WT (WTC-11) kidney organoids. In contrast to the human kidney, in which nephrons comprise only a small portion of the whole kidney, most of a kidney organoid is comprised of tubular structures [23], and it was reported that loss of NCCT leads to major structural remodeling of the renal distal tubule that goes along with marked changes in glomerular and tubular function, along with significant structural changes of the distal nephron, such as early DCT atrophy and CNT hypertrophy, as shown in an experiment on a mouse model lacking the expression of the NCCT gene (NCCT^−/−^ mice) [24]. Therefore, it is possible that NCCT defects may interrupt the formation of tubular structure in kidney organoids, including decreased expression of PODXL, LTL, and ECAD, and, finally, they may result in less maturation of GIT kidney organoids. Furthermore, correction of the *SLC12A3* gene recovered not only NCCT but also the expression of PODXL, LTL and ECAD, and the maturity of kidney organoids. Nevertheless, further study is needed to determine the mechanisms for the morphological and functional defects of the nephron following downregulation of the *SLC12A3* gene. For example, ion movement with a fluorescent indicator (e.g., CoroNa Green) or patch-clamp tool may be helpful in understanding of the NCCT channel function.

Interestingly, even though only one mutation (c. 47_48del in exon 1) out of two pathogenic mutations in this hiPSC was successfully corrected by CRISPR/Cas9, expression of NCCT and other tubule markers was successfully rescued. It can be accounted for by the previous knowledge that most GIT patients are confirmed to have biallelic mutation in the *SLC12A3* gene and showed compound heterozygous variants and single mutation, which can induce clinical manifestation, and is very rare [25,26]. In this study, gene mutation status changed from two pathogenic mutations to a single mutation after gene correction, which may be enough to rescue the GIT phenotype. With the same context, our study results can be used to verify the significance of the mutation. There are still many mutations of which pathologic significance has not been clarified [27]. The clinical significance has been described in case reports or genetic studies with animal models [28]. Using our platform, we can make patient-derived hiPSCs, differentiate them into kidney organoids, and investigate whether the mutation is associated with the disease phenotype. Furthermore, as we did in this study, we can see whether correction of a mutation can rescue the disease phenotype; then, we can confirm its significance. We suggest that our platform can be utilized as the platform to investigate the role of genetic mutations that are able to be translated into other genetic disorders.

Our study has some limitations to be solved. First, most genetic disorders, including GIT, are not monogenic; hence, direct gene correction would be difficult to translate to all patients uniformly. Alternatively, targeted disruption or insertion of other specific genes associated with disease development or protection can be applied to all patients universally, irrespective of mutation type [29,30]. Therefore, the strategy to knock in specific genes to recover the expression of NCCT can be explored. Next, an effective method to deliver CRISPR therapeutics to the kidney should be devised. The adenovirus vector-based approach is proposed as an ideal platform for CRISPR/Cas9 delivery [31,32,33]; however, there have been no attempts or evidence surrounding the use of adenovirus vector infection of kidney epithelial cells. Further investigation may be required to overcome these issues.

Taken together, kidney organoids from GIT patient-derived hiPSCs recapitulate the disease phenotype in terms of decreased expression of NCCT, and CRISPR/Cas9-mediated correction of *SLC12A3* gene mutation rescue the disease phenotype. Our results suggest that CRISPR/Cas9-mediated gene correction can be proposed as a new therapeutic strategy for the treatment of GIT.

## 4. Methods and Methods

### 4.1. hiPSC Generation and Cell Culture

GIT patient hiPSCs were generated using peripheral blood samples as described in a previous study (CMCi002-A [alternate name, CMC-GIT-001]) [7]. *SLC12A3* gene sequencing revealed heterozygous c. 47_48del and c.2963 T > C mutations. The PBMC obtained from patients was cultured for 4 days at 37 °C in an incubator with 5% CO_2_ in StemSpan medium (09650; STEMCELL Technologies, Vancouver, Canada), which includes StemSpan CC100 (02690; STEMCELL Technologies) to expand CD34-positive cells. The expanded PBMCs were transfected using the CytoTune-iPS Sendai Reprogramming Kit (A16517; Life Technologies, Carlsbad, CA, USA), which includes the Yamanaka factors (Oct4, Sox2, KLF4, and c-Myc). PBMCs were induced to form hiPSCs via centrifugation, and the resulting attached cells were expanded and purified using colony picking.

### 4.2. Construction of sgRNA and ssODN

Single guide RNA (sgRNA) and ssODN for gene correction in the target regions were designed using the ToolGen CRISPR/Cas9 technology service. sgRNA are listed as follows: 5′-GTGAAGCGCCCGCTGCAAAGTGG-3′ (reverse frame) for exon 1. ssODN are listed as follows: 5′-TGGACACCCAGGCGACAATGGCAGAACTGCCCACAACAGAGACGCCTGGGGACGCCACTTTGTGCAGCGGGCGCTTCACCATCAGCACACTGCTGAGCAGTGATGAGCCCTCTCCACCAGCTGCCTATGAC-3′ for exon 1. We designed an ssODN donor with the *ACCIII* enzyme site insert for exon 1.

### 4.3. In Vitro Cas9 Cleavage Assay

To check sgRNA efficiency in GIT patient hiPSCs, we mixed the following components: 250 ng of sgRNA, 500 ng of Cas9 protein (ATS-0010, Bioneer Daedeok-gu, Daejeon), 3 μL of genomic hiPSC DNA, and reaction buffer, then incubated the solution at 37 °C for 1.5 h. A total of 1μL of RNase was incubated for a further 20 min at 37 °C, followed by the addition of stop solution. The efficiency of Cas9-mediated cleavage was analyzed through the fragmented PCR products obtained using gel electrophoresis using primer sets (F-AAGAGCCACTCCAGGACTCA, R-GAGGAAGGAGTGCAGGTCAG for Exon 1 region).

### 4.4. hiPSC Transfection for Gene Editing

sgRNA and Cas9 protein were introduced by electroporation using NEPAGENE (NEPA21, Nepa Gene Co., Ltd. Chiba, Japan) as the ribonuleoprotein (RNP) complex. Briefly, the RNP complex was formed by mixing 30 μg Cas9 protein with 25 μg sgRNA and incubating the mixture at room temperature for 10 min. These were then electroporated to 1 × 10^6^ hiPSC with 100μL of Opti-MEM Medium (31985062; Thermo Fisher Scientific, Grand Island, NY, USA). At least 72 h after transfection, genomic DNA (gDNA) was extracted using the Genomic DNA Extraction Kit (MN740230.250; Macherey-Nagel, Allentown, PA, USA) according to the manufacturer’s protocol. To identify the gene editing efficiency and exact sequence edited, we utilized the IN/DEL analysis service for CRISPR validation (ATC-0120, Bioneer). In addition, correction of targeted genes was confirmed by *AccIII* restriction digestion of PCR products before and after gene correction.

### 4.5. Flow Cytometry

hiPSCs cells were dissociated using TE (15400054; Life Technologies). The cells were washed twice with FACS buffer (phosphate-buffered saline [PBS] containing 1% bovine serum albumin and 10 mM sodium azide), permeabilized for 30 min using flow cytometry fixation and permeabilization solution (554714; BD Biosciences, San Jose, CA, USA), washed with wash buffer, and stained with stage-specific embryonic antigens SSEA-4 (813-70, 1:100; Santa Cruz Biotechnology, Dallas TX, USA) or TRA-1–81 (TRA-1-81, 1:100; Santa Cruz Biotechnology) surface antibodies for 30 min. Intracellular staining for NANOG (1E6C4, 1:100; Santa Cruz Biotechnology) was performed by sequential incubations with fixation and permeabilization solutions (A and B Fix & Perm Solutions, Invitrogen, BD Bioscience). These cells were incubated with FITC-conjugated secondary antibody (BD Bioscience). IgG1 isotypes, control to FITC (NANOG) cells, were analyzed using a FACS Canto II flow cytometer (BD Biosciences). The data were analyzed using FlowJo software version 10.8.1. (Becton, Dickinson & Company, Ashland, OR, USA).

### 4.6. Immunofluorescence

hiPSCs or kidney organoids were washed once with phosphate-buffered saline (PBS), fixed with 4% paraformaldehyde for 10 min at 4 °C, and were blocked in 5% donkey serum in PBS-T (0.3% Triton X-100 in PBS) for 1 h at room temperature (RT). hiPSCs or kidney organoids were incubated with primary antibodies, anti-NANOG (1E6C4, 1:100; Santa Cruz Biotechnology), anti-SSEA4 (MAB4304, 1:100; Millipore Sigma, Burlington, MA, USA), anti–TRA-1-81 (MAB4381, 1:100; Millipore Sigma), anti-PAX6 (sc-81649, 1:20; Santa Cruz Biotechnology), anti-SM22A (sc-53932, 1:40; Santa Cruz Biotechnology), anti-FOXA2 (sc-374376, 1:50; Santa Cruz Biotechnology), biotinylated Lotus Lectin (LTL, B-1323,1:100; Vector Laboratories, Burlingame, CA, USA), E-cadherin (ECAD, ab11512, 1:50; abcam, Cambridge, UK), podocalyxin (PODXL, BAF1658, 1:100; R&D Systems, Minneapolis, MN USA), and anti-NCC (ab224762, 1:50; abcam) overnight at 4 °C. Then, the cells were stained with secondary antibodies; Alexa Fluor 488-donkey anti-mouse IgG (A32766, 1:250; Invitrogen, Camarillo, CA, USA), Alexa Fluor 647-donkey anti-goat IgG (A32849, 1:250; Invitrogen), Cyanine3(Cy3)-streptavidin (016-160-084, 1:1000; Jackson ImmunoResearch, West Grove, PA, USA), Alexa Fluor 488-donkey anti-rat IgG (A48269, 1:250; Invitrogen), and Cy3-conjugated donkey anti-rabbit (711-165-152, 1:1000; Jackson ImmunoResearch). Nucleic acid staining was performed by incubation with 4′,6-diamidine-2-phenylindole (10236276001, DAPI, 1:5000; Roche, Basel, Switzerland) for 30 min at room temperature. Images were obtained using a Zeiss LSM700 confocal microscope (Carl Zeiss MicroImaging GmbH, Jena, Germany).

### 4.7. Tri-Lineage Differentiation

Regarding trilineage differentiation, a StemMACS™ Trilineage Differentiation Kit (Miltenyi Biotec, Gaithersburg, MD, USA) was used. Putative hiPSCs were cultivated for seven days in three different chemically defined media, driving the differentiation into three germ layers. The three differentiated germ layers were fixed in 4% paraformaldehyde and stained with the following antibodies: anti-PAX6 antibody for the ectoderm, anti-SM22A antibody for the mesoderm, and anti-FOXA2 antibody for the endoderm.

### 4.8. Karyotype Analysis

Pluripotent cells were cultured in culture plates coated with Matrigel in conditioned media for 3~5 days. The cells were transported to GenDix Research Center (GenDix Inc., Seoul, Republic of Korea), where cell harvest and karyotype analysis of metaphase chromosomes was performed using G-banding.

### 4.9. Mycoplasma Detection

Mycoplasma analysis was performed using the e-Myco VALiD Mycoplasma PCR kit (iNtRON Biotechnology) and was absent in all cases.

### 4.10. Kidney Organoid Differentiation from hiPSCs

Healthy control (WTC-11), GITpatient, and GITpatient^corr^ hiPSCs were differentiated into kidney organoids following the previously published protocol [10,11,12,14] (Figure 3B). Briefly, hiPSCs were plated in mTeSR1 medium (05850; STEMCELL Technologies Vancouver, Canada) supplemented with 10 μM Y-27632 (1293823; Biogems, Westlake Village, CA, USA) onto 24-well plates pre-coated with 1.25% Corning Matrigel^®^ hESC Qualified Matrix. After 24 h, the medium was exchanged with 2.5% of Matrigel^®^ in mTeSR1. On the fourth day, the medium was replaced with Advanced RPMI (1263302; Thermo Fisher Scientific, Grand Island, NY, USA) supplemented with 12 µM CHIR-99021 (STEMCELL Technologies). Approximately 36 h later, the medium was changed to Advanced RPMI with B27 supplement (17504044, Thermo Fisher Scientific). Organoids were cultured in this medium until collection on day 21.

### 4.11. qRT-PCR

Total RNA was extracted from cells using RNA-Bee (CD-105B; Tel-Test, Friendswood, TX, USA) as per the manufacturer’s instructions. Around 1 μg of total RNA was employed for cDNA synthesis with a cDNA synthesis kit (DYRT1120; Dyne Bio Inc, Seongnam-si, South Korea). RT-qPCR was performed using SYBR Green Master Mix (DYRT1200; Dyne Bio Inc) on a LightCycler 480 system (Roche, Basel, Switzerland) using a primer set for NCC or GAPDH (995′-CTCCACAGCCAGGCAAATA-3′ and 5′-GACCTGAGATCATGCCATTACA-3′ for human NCC; 5′-TCG GAG GAC TTG AAT GGA ATG-3′ and 5′-TCG CCA TTT CCC GTT TCT-3′ for human GAPDH). All data were normalized against GAPDH and were calculated using the ΔΔCq method.

### 4.12. Immunoblot Analysis

Kidney organoids were washed with PBS and lysed with 10mM Tris (pH 7.5) containing 1% sodium dodecyl sulfate (SDS) and 1 mM NaVO_4_. Equal amounts of protein were separated on 7% SDS-PAGE gels and transferred to a polyvinylidene difluoride membrane (IB401001; Invitrogen, Carlsbad, CA, USA) using the iBlot2 Gel Transfer Device (IB21001; Invitrogen). Membranes were blocked with 5% skim milk in PBS with 0.1% Tween 20 detergent (T-PBS) for 1 h at room temperature and incubated with anti-NCC (ab224762, 1:500; abcam) and biotinylated Lotus Lectin (LTL, B-1323,1:100; Vector Laboratories, Burlingame, CA, USA), E-cadherin (ECAD, 610405, 1:500; BD Biosciences, San Jose, CA, USA), and b-actin (1:2000, 3700; Cell signaling Technology, Danvers, MA, USA) at 4 °C overnight. Membranes were then washed 3 times with T-PBS and incubated for 1 h with a goat anti-rabbit IgG-HRP conjugate (1706515, 1:1000; Bio-Rad, Hercules, CA, USA), a goat anti-mouse IgG-HRP conjugate (1706516, 1:1000; Bio-Rad), and a Strep Tactin-HRP conjugate (1610381, 1:1000; Bio-Rad). After 4 washes with T-PBS, the protein of interest was detected using an enhanced chemiluminescence system (WSE-7110, ATTO Corp., Tokyo, Japan). Quantification of relative densities was performed with the control group set at 100%; densities were normalized to those of b-actin bands from the same gel (Quantity One version 4.4.0; Bio-Rad).

### 4.13. Statistical Analyses

Data have been expressed in terms of mean ± standard error (SE) from at least three independent experiments. Multiple comparisons between groups were performed by one-way analysis of variance with the Bonferroni post hoc test using Prism software (version 7.03 for Windows; GraphPad, La Jolla, CA, USA). Statistical significance was set at *p* < 0.05.

## Figures and Tables

**Figure 1 ijms-24-03019-f001:**
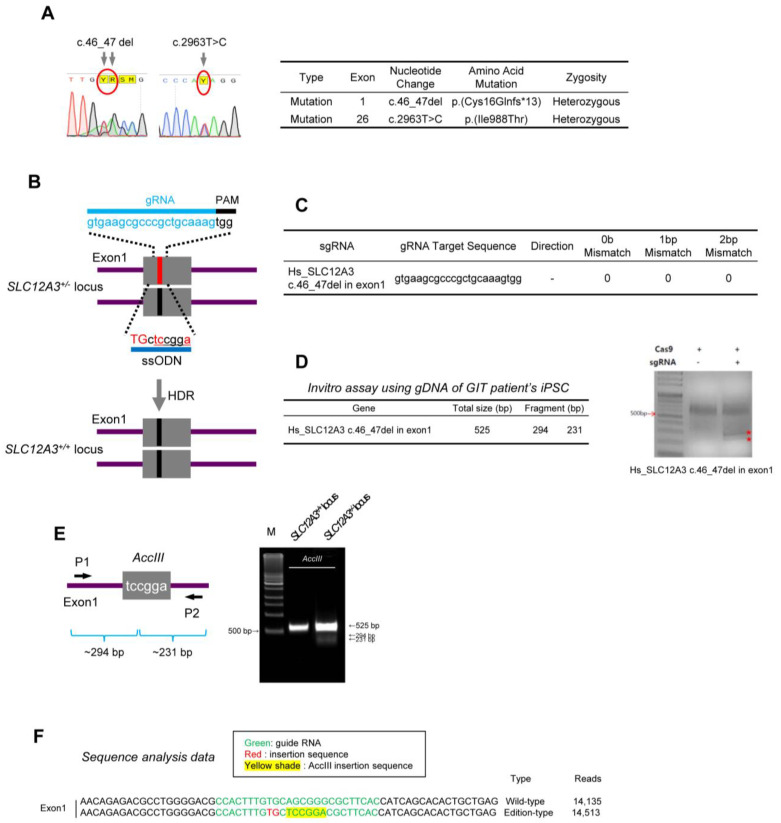
Establishment of CRISPR/Cas9 RNP-mediated SLC12A3 gene editing of CMC-GIT-001 iPSCs. (**A**) Sequencing analysis showing *SLC12A3* gene mutation. (**B**) Strategy for correcting *SLC12A3* mutation. The sequence of gRNA is shown with the PAM sequence. The red line represents the mutant allele, and the blue line represents the wild-type allele. HDR, homology-directed repair; ssODN, single-stranded oligodeoxynucleotide. The red uppercase letters highlight the corrected base. (**C**) sgRNA target site sequence. (**D**) In vitro assay result using gDNA from GIT patient hiPSCs. (**E**) PCR validation using digestion of PCR products before and after gene correction in either exon 1 by *AccIII* enzyme. (**F**) Confirmation of *SLC12A3* gene correction in CMC-GIT-001 hiPSCs by DNA sequencing. Enzyme sites are highlighted with a yellow shade. Corrected sequences are highlighted in red.

**Figure 2 ijms-24-03019-f002:**
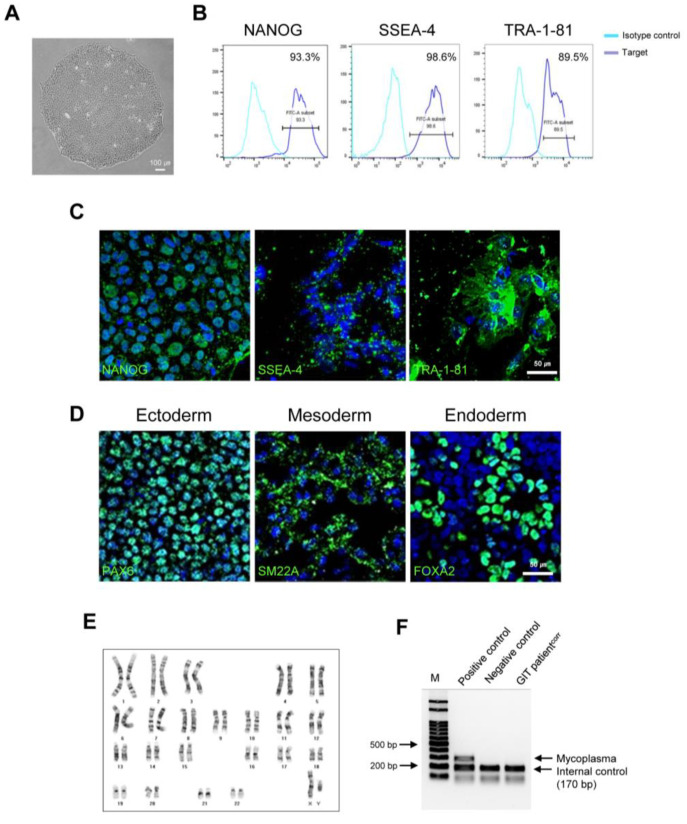
Characterization of corrected iPSCs from GIT patients (CMC-GIT-001^corr^). (**A**) Morphology of corrected GIT patient iPSCs (CMC-GIT-001^corr^). (**B**) Flow cytometry analysis of cells expressing NANOG, SSEA-4, and TRA-1-81. (**C**) Representative immunofluorescence image of pluripotency markers NANOG, SSEA-4, and TRA-1-81. (**D**) Immunofluorescence staining of three germ layer markers. Ectoderm, mesoderm, and entoderm differentiation were detected by PAX4, SM22a, and FOX2A, respectively. Scale bar = 50 μm. (**E**) Chromosome karyotyping of CMC-GIT-001^corr^ hiPSCs. (**F**) Mycoplasma detection by PCR, negative.

**Figure 3 ijms-24-03019-f003:**
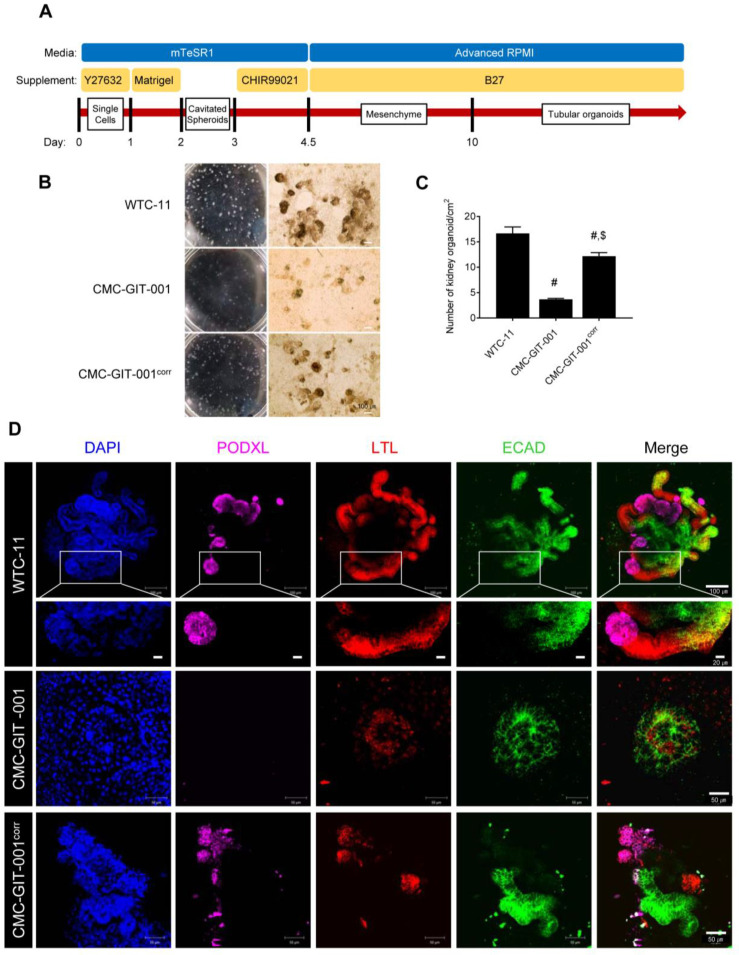
Differentiation of kidney organoids from CMC-GIT-001 and CMC-GIT-001^corr^ hiPSCs. (**A**) Schematic timeline of the hiPSC differentiation protocol. (**B**) Whole-well bright field observations and light micrograph of the kidney organoids. Scale bar = 100. (**C**) Comparison of the number of matured kidney organoids in CMC-GIT-001 and CMC-GIT-001^corr^. Data are presented as mean ± SE. # *p* < 0.05 vs. WTC-11; $ *p* < 0.05 vs. CMC-GIT-001. (**D**) Representative immunofluorescence image of PODXL, LTL, and ECAD kidney organoids from WTC-11 as a control, CMC-GIT-001, and CMC-GIT-001^corr^. Scale bar = 50 or 100 μm.

**Figure 4 ijms-24-03019-f004:**
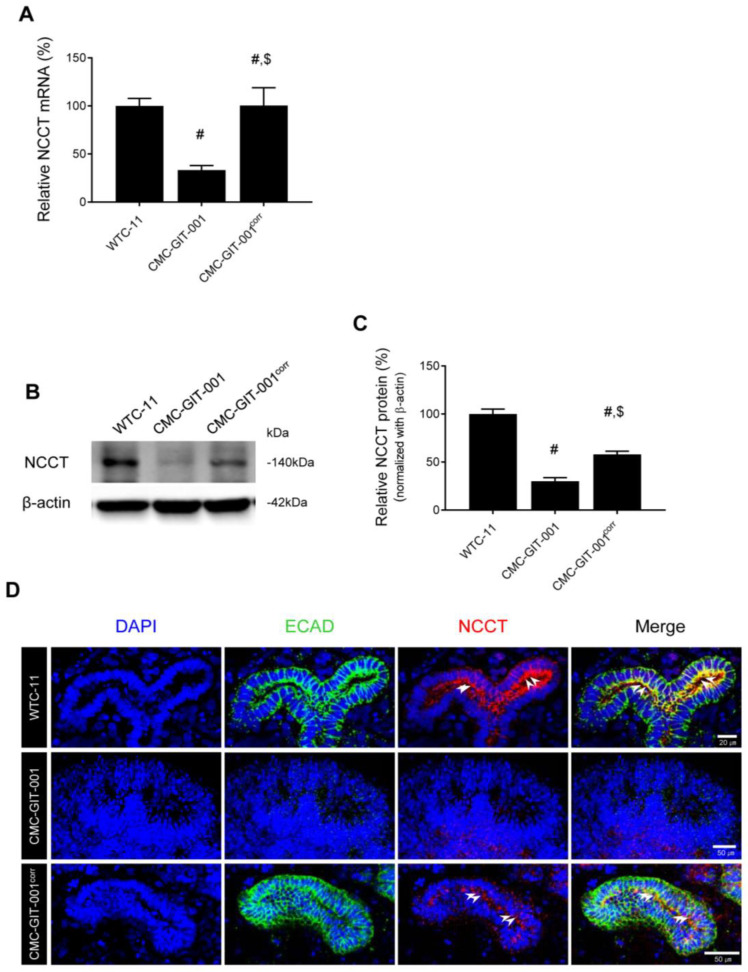
Expression of NCCT in kidney organoids from CMC-GIT-001 and CMC-GIT-001^corr^ iPSCs. The NCCT mRNA (**A**) and protein expression (**B**,**C**) in the kidney organoids from WTC-11 as a control, CMC-GIT-001, and CMC-GIT-001^corr^. Relative NCCT mRNA and protein expression was presented after normalization to GAPDH and β-actin expression, respectively. (**D**) Representative immunofluorescence image of ECAD and NCCT kidney organoids from WTC-11 as a control, CMC-GIT-001, and CMC-GIT-001^corr^. Data are presented as mean ± SE. # *p* < 0.05 vs. WTC-11; $ *p* < 0.05 vs. CMC-GIT-001. # *p* < 0.05 vs. WTC-11; $ *p* < 0.05 vs. CMC-GIT-001. Scale bar = 20 or 50 μm.

**Figure 5 ijms-24-03019-f005:**
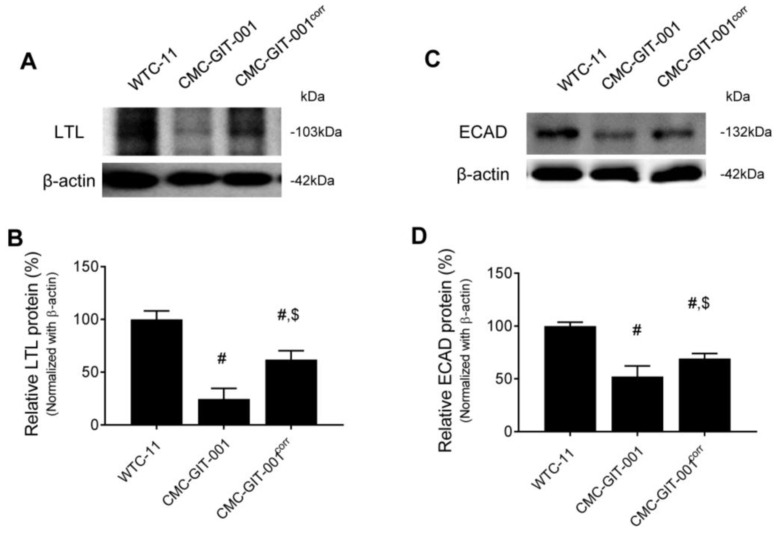
Expression of LTL and ECAD in kidney organoids from CMC-GIT-001 and CMC-GIT-001^corr^ iPSCs. Representative immunoblot image and LTL (**A**,**B**) and ECAD (**C**,**D**) quantification in the kidney organoids from WTC-11 as a control, CMC-GIT-001, and CMC-GIT-001^corr^. The relative optical densities of the bands in each lane were normalized to those of each band of β-actin. Data are presented as mean ± SE. # *p* < 0.05 vs. WTC-11; $ *p* < 0.05 vs. CMC-GIT-001.

## Data Availability

All datasets generated for this study are included in the article.

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
