# Peer review of "CRISPR-Cas9-Mediated Correction of *SLC12A3* Gene Mutation Rescues the Gitelman’s Disease Phenotype in a Patient-Derived Kidney Organoid System"

_ijms, 2023, doi:10.3390/ijms24033019_

Round 1

Reviewer 1 Report

I have read with great interest this manuscript assessing the role of gene editing for the correction of SLC12A3 mutation. The Authors applied a CRISPR/Cas9 system to correct the gene mutation on patient-derived hiPSCs and tested the recovered phenotype on hiPSCs-derived kidney organoids.

The following are my comments.

MAJOR

- Can you please provide a better quality image for figure 3B? It is really difficult for the readers to understand the morphology and architecture from these pictures.

- This also applies to figure 3D. Moreover, in these IF images, can the authors provide more details of the differential expression and position of any nephron marker?

- Can the other provide any functional test to assess the NCCT restored function?

- Can the authors comment on the reduced expression of NCCT protein compared to the restored mRNA expression in CMC-GIT-001-corr organoids?

- Did the authors check PODXL expression on kidney organoids?

MINOR

Figure 2 F – Please specify the bp for the internal control.

Abstract. The sentence “hiPSCs were successfully differatianted” is written twice.

Author Response

Reviewer 1:

I have read with great interest this manuscript assessing the role of gene editing for the correction of SLC12A3 mutation. The Authors applied a CRISPR/Cas9 system to correct the gene mutation on patient-derived hiPSCs and tested the recovered phenotype on hiPSCs-derived kidney organoids.

<Major Comments>

  1. Can you please provide a better quality image for figure 3B? It is really difficult for the readers to understand the morphology and architecture from these pictures

Response: We changed the figure 3B with a high resolution image as reviewer’s suggestion.

  1. This also applies to figure 3D. Moreover, in these IF images, can the authors provide more details of the differential expression and position of any nephron marker?

Response: We added high magnification insets in figure 3D.

  1. Can the other provide any functional test to assess the NCCT restored function?

Response: Since the SLC12A3 gene encode sodium/chloride cotransporter in distal tubule, we agree with review's point out the appropriate to investigate the reabsorption of ion such as sodium and chloride, which are functional parts in addition to the expression of RNA and protein level of NCCT. Ion movement with a fluorescent indicator (e.g. CoroNa Green) or patch-clamp tool may helpful to understanding of the NCCT channel function, however there is a limited to performing physiological condition like in vivo using organoids within culture dish.

  1. Can the authors comment on the reduced expression of NCCT protein compared to the restored mRNA expression in CMC-GIT-001-corr organoids?

Response: As the reviewer pointed out, the NCCT mRNA and protein in the CMC-GIT-001corr group showed 3-fold and 2-fold increase, respectively, compared to CMC-GIT-001 group. We cannot clearly response in this study however, many research paper shows similar expression patterns that may not be the same in the expression of mRNA and protein like in this study. We carefully assume that this is one of reasons, but it is necessary to investigate the periodic expression of mRNA and protein of NCCT during kidney organoid differentiation.

  1. Did the authors check PODXL expression on kidney organoids?

Response: Sure, we checked the PODXL expression on kidney organoids as represented in figure 3D. Unfortunately, kidney organoid of gitelman patient did not stain with PODXL antibody. We cannot clearly answer regarding this, but loss of NCCT gene leads a retarded differentiation (e.g. morphological immature, low number) in the patient kidney organoid as we mentioned in the discussion section. Therefore, it is possible that NCCT defects may interrupt the formation of tubular structure in kidney organoids, including decreased expression of PODXL.

<Minor Comments>

  1. Figure 2 F – Please specify the bp for the internal control.

Response: We specify the bp for the internal control as 170bp in figure 2F.

  1. Abstract. The sentence “hiPSCs were successfully differatianted” is written twice.

Response: We deleted repeated sentence in the abstract.

Author Response

Reviewer #2:

The study was aimed to test the potential contribute of gene correction using CRISPR/Cas9 in the rescue of the disease phenotype of Gitelman’s disease (GIT). Specifically, the authors used human induced pluripotent stem cell (hiPSC)-derived kidney organoids to model GIT. The major conclusions were that renal organoids from GIT patient-derived hiPSC recapitulated the Gitelman’s disease phenotype, and the correction of SLC12A3 mutation utilizing CRISPR-Cas9 technology provided therapeutic insight My comments are:

  1. Introduction. Line 56. hiPSC was cited for the first time in the main text. I suggest to the authors explain the meaning of acronym.

Response: We added abbreviation of hiPSC as human induced pluripotent stem cell.

  1. In the results section the resolution of figures and panels must be improved.

Response: We revised the figure with high resolution and uploaded original files.

Round 2

Reviewer 1 Report

Thanks for the clarifications to the questions I have raised, nonetheless these points should be also discussed  within the manuscript.

This applies to the following points of my previous review

  1. Can the other provide any functional test to assess the NCCT restored function?
  2. Can the authors comment on the reduced expression of NCCT protein compared to the restored mRNA expression in CMC-GIT-001-corr organoids?
  3. Did the authors check PODXL expression on kidney organoids?

Author Response

Reviewer 1:

Thanks for the clarifications to the questions I have raised, nonetheless these points should be also discussed within the manuscript.

This applies to the following points of my previous review

Response: We really appreciated your precise revision. We appropriately added the responses in the discussion section (marked with red color).

  1. Can the other provide any functional test to assess the NCCT restored function?

Response: Since the SLC12A3 gene encode sodium/chloride cotransporter in distal tubule, we agree with review's point out the appropriate to investigate the reabsorption of ion such as sodium and chloride, which are functional parts in addition to the expression of RNA and protein level of NCCT. Ion movement with a fluorescent indicator (e.g. CoroNa Green) or patch-clamp tool may helpful to understanding of the NCCT channel function, however there is a limited to performing physiological condition like in vivo using organoids within culture dish.

We addressed this issue in the discussion section as follow;

Interestingly, the proportion of mature kidney organoids was significantly low, and tubular proteins, such as LTL and ECAD, showed decreased expression in CMC-GIT-001 kidney organoids in comparison with WT (WTC-11) kidney organoids. In contrast to the human kidney, in which nephrons comprise only a small portion of the whole kidney, most of a kidney organoid is comprised of tubular structures [23], and it was reported that loss of NCCT leads to major structural remodeling of the renal distal tubule that goes along with marked changes in glomerular and tubular function, along with significant structural changes of the distal nephron, such as early DCT atrophy and CNT hypertrophy, as shown in an experiment on a mouse model lacking the expression of the NCCT gene (NCCT¯/¯ mice) [24]. Therefore, it is possible that NCCT defects may interrupt the formation of tubular structure in kidney organoids, including decreased expression of LTL and ECAD, and, finally, they may result in less maturation of GIT kidney organoids. Furthermore, correction of the SLC12A3 gene recovered not only NCCT but also the expression of LTL and ECAD and the maturity of kidney organoids. Nevertheless, further study is needed to determine the mechanisms for the morphological and functional defects of the nephron following downregulation of the SLC12A3 gene. For example, ion movement with a fluorescent indicator (e.g. CoroNa Green) or patch-clamp tool may helpful to understanding of the NCCT channel function.

  1. Can the authors comment on the reduced expression of NCCT protein compared to the restored mRNA expression in CMC-GIT-001-corr organoids?

Response: As the reviewer pointed out, the NCCT mRNA and protein in the CMC-GIT-001corr group showed 3-fold and 2-fold increase, respectively, compared to CMC-GIT-001 group. We cannot clearly response in this study however, many research paper shows similar expression patterns that may not be the same in the expression of mRNA and protein like in this study. We carefully assume that this is one of reasons, but it is necessary to investigate the periodic expression of mRNA and protein of NCCT during kidney organoid differentiation.

We addressed this issue in the discussion section as follow;

Next, we tried to correct for SLC12A3 gene mutation using CRISPR/Cas9. Some previous studies already suggest that CRISPR/Cas9-mediated gene correction has potential as a therapeutic strategy for genetic disease, especially when an appropriate treatment strategy is unavailable, such as GIT [15,19]. For example, correction of mutation in hereditary hematologic disorders, such as ß-thalassemia or hemophilia A, using patient-derived hiPSCs rescues the disease phenotype [20,21]. In another study, CRISPR/Cas9-mediated correction was performed in dominant optic atrophy in patient-derived hiPSCs [22]. In this study, CMC-GIT-001 showed heterozygote mutations in two different sites reported as pathogenic for the development of GIT according to ACMG guidelines. After gene correction by CRISPR/Cas9, mutation of c. 47_48del in exon 1 was successfully corrected, and mRNA and protein of NCCT were also significantly recovered according to qRT-PCR and immunoblot analysis. This finding suggests the successful rescue of GIT disease phenotype through CRISPR/Cas9. However, we found that NCCT, the expression of mRNA and protein recovered by 3-fold and 2-fold, respectively compared with CMC-GIT-001 according to qRT-PCR and immunoblot analysis. It is necessary to investigate the periodic expression of mRNA and protein of NCCT during kidney organoid differentiation in the future.

  1. Did the authors check PODXL expression on kidney organoids?

Response: Sure, we checked the PODXL expression on kidney organoids as represented in figure 3D. Unfortunately, kidney organoid of gitelman patient did not stain with PODXL antibody. We cannot clearly answer regarding this, but loss of NCCT gene leads a retarded differentiation (e.g. morphological immature, low number) in the patient kidney organoid as we mentioned in the discussion section. Therefore, it is possible that NCCT defects may interrupt the formation of tubular structure in kidney organoids, including decreased expression of PODXL.

We addressed this issue in the discussion section as follow;

Interestingly, the proportion of mature kidney organoids was significantly low, and tubular proteins, such as LTL and ECAD showed decreased expression and PODXL did not detected in CMC-GIT-001 kidney organoids in comparison with WT (WTC-11) kidney organoids. In contrast to the human kidney, in which nephrons comprise only a small portion of the whole kidney, most of a kidney organoid is comprised of tubular structures [23], and it was reported that loss of NCCT leads to major structural remodeling of the renal distal tubule that goes along with marked changes in glomerular and tubular function, along with significant structural changes of the distal nephron, such as early DCT atrophy and CNT hypertrophy, as shown in an experiment on a mouse model lacking the expression of the NCCT gene (NCCT¯/¯ mice) [24]. Therefore, it is possible that NCCT defects may interrupt the formation of tubular structure in kidney organoids, including decreased expression of PODXL, LTL and ECAD, and, finally, they may result in less maturation of GIT kidney organoids. Furthermore, correction of the SLC12A3 gene recovered not only NCCT but also the expression of PODXL, LTL and ECAD and the maturity of kidney organoids. Nevertheless, further study is needed to determine the mechanisms for the morphological and functional defects of the nephron following downregulation of the SLC12A3 gene. For example, ion movement with a fluorescent indicator (e.g. CoroNa Green) or patch-clamp tool may helpful to understanding of the NCCT channel function.
